# Marine Morbilliviruses: Diversity and Interaction with Signaling Lymphocyte Activation Molecules

**DOI:** 10.3390/v11070606

**Published:** 2019-07-03

**Authors:** Kazue Ohishi, Tadashi Maruyama, Fumio Seki, Makoto Takeda

**Affiliations:** 1Faculty of Engineering, Tokyo Polytechnic University, 1583, Iiyama, Atsugi, Kanagawa 243-0297, Japan; 2School of Marine Biosciences, Kitasato University, 1-15-1, Kitazato, Minami, Sagamihara, Kanagawa 252-0373, Japan; 3Department of Virology III, National Institute of Infectious Diseases, 4-7-1, Gakuen, Musashimurayama, Tokyo 208-0011, Japan

**Keywords:** cetacean morbillivirus, host specificity, marine mammal, morbillivirus, phocine distemper virus, receptor, signaling lymphocyte activation molecule

## Abstract

Epidemiological reports of phocine distemper virus (PDV) and cetacean morbillivirus (CeMV) have accumulated since their discovery nearly 30 years ago. In this review, we focus on the interaction between these marine morbilliviruses and their major cellular receptor, the signaling lymphocyte activation molecule (SLAM). The three-dimensional crystal structure and homology models of SLAMs have demonstrated that 35 residues are important for binding to the morbillivirus hemagglutinin (H) protein and contribute to viral tropism. These 35 residues are essentially conserved among pinnipeds and highly conserved among the Caniformia, suggesting that PDV can infect these animals, but are less conserved among cetaceans. Because CeMV can infect various cetacean species, including toothed and baleen whales, the CeMV-H protein is postulated to have broader specificity to accommodate more divergent SLAM interfaces and may enable the virus to infect seals. In silico analysis of viral H protein and SLAM indicates that each residue of the H protein interacts with multiple residues of SLAM and vice versa. The integration of epidemiological, virological, structural, and computational studies should provide deeper insight into host specificity and switching of marine morbilliviruses.

## 1. Introduction

Morbilliviruses belong to the genus *Morbillivirus* and the family Paramyxoviridae. *Morbillivirus* species have high host specificity and infection results in a highly contagious and devastating disease [1]. Four classic *Morbillivirus* species are well-known in terrestrial mammals: measles virus (MV; measles morbillivirus) in humans, rinderpest virus (RPV; rinderpest morbillivirus) in cattle, peste des petits ruminants virus (PPRV; small ruminant morbillivirus) in goats and sheep, and canine distemper virus (CDV; canine morbillivirus) in dogs [1]. Two marine morbillivirus species were isolated during the 1980s and 1990s from dead seals or cetaceans from mass-die-offs or strandings, and named phocine distemper virus (PDV; phocine morbillivirus) and cetacean morbillivirus (CeMV) [2,3,4,5,6]. In addition to these six well-known species, novel morbilliviruses have been identified in domestic cats and bats [7,8]. In this review, we describe the six well-known viruses with a focus on the two marine morbilliviruses, PDV and CeMV.

Morbilliviruses are lymphotropic and induce severe systemic disease with high morbidity and mortality. Morbillivirus infection often causes immunosuppression leading to secondary or opportunistic infections in the host [9,10,11]. Rinderpest caused by RPV represents a significant threat to livestock. However, using an effective vaccine, the Global Rinderpest Eradication Program was successful, with the Food and Agriculture Organization (FAO) and World Organization for Animal Health (OIE) declaring the global eradication of this virus in 2011. Following the success of this eradication program, the FAO and OIE have set a goal of global PPRV eradication by 2030 using the same strategy. MV causes high child mortality rates in developing countries, and, currently, the World Health Organization is promoting a MV elimination program through vaccination.

Morbilliviruses have single negative-stranded RNA genomes encoding eight viral proteins: the nucleocapsid protein (N), phosphoprotein (P), two non-structural virulence factors (C and V), matrix protein (M), fusion protein (F), hemagglutinin protein (H), and RNA polymerase (L) [1]. The two viral surface glycoproteins, H and F, play essential roles in viral entry into host cells. The H protein is required for viral attachment to receptors on host cells during the initial stages of infection. Receptor binding then triggers a conformational change in the F protein, inducing fusion of viral and host cell membranes and enabling viral entry into the host cell [12,13,14,15]. Thus, viral binding to a cellular receptor plays a key role in determining tropism and host specificity. Morbilliviruses use two cellular receptors, signaling lymphocyte activation molecule (SLAM) [16] and nectin-4 [17,18]. These receptors belong to the immunoglobulin superfamily and are expressed on immune cells and epithelial cells, respectively. SLAM functions during initial infection and during viral spread throughout the body, whereas nectin-4 plays a role during the last step of disease (virion release from epithelial cells) [17,19,20,21,22,23,24,25,26,27,28]. The amino acid sequences of SLAMs are more divergent among host species [29,30] compared with those of nectin-4, which are highly conserved [31,32]. This suggests that SLAM plays a greater role than nectin-4 in determining the host specificity of morbilliviruses. Modulating interactions between viral H proteins and SLAMs may alter virus–host specificity and host range.

Among marine mammals, cetaceans are phylogenetically classified within the order Cetartiodactyla together with the Artiodactyla, which includes cow and sheep. Cetaceans are divided into toothed whales (Odontoceti), including bottlenose and striped dolphins, and baleen whales (Mysticeti), including fin and minke whales. Cetaceans live solely in aquatic environments, and sirenians (order Sirenia) such as manatees and dugongs also live in this environment. This unique niche ensures geographic isolation from land animals. Pinnipeds, including seals and walruses, belong to the order Carnivora and have not completely adapted to aquatic environments with the birth of their young and raising of newborns taking place on land. This unique ecological niche is epidemiologically important as a vector linking aquatic and land environments.

The number of reported PDV and CeMV infections of various marine mammal species in oceans worldwide continues to rise. For transmission of the virus from species to species, ecological factors such as habitat areas, migration patterns, and animal behavior are crucial. Animal population size and density are also key elements for viral maintenance among animal populations. The interactions between viral components and host molecules such as receptors or innate immune factors are another important factor in host–pathogen interaction. In this review, recent epidemiological findings of marine morbilliviruses are described from the viewpoint of the interaction between viruses and their major host cell receptor, SLAM.

## 2. Discovery of Marine Morbilliviruses

In the period 1988–1989, approximately 18,000 harbor seals (*Phoca vitulina*) and several hundred grey seals (*Halichoerus grypus*) died on the northern European coastline. Distemper-like gross symptoms suggested that a morbillivirus was the causative agent [33]. Detailed serological, virological, and immunohistochemical examination showed that the agent was a new member of the genus *Morbillivirus,* named PDV [34,35,36]. The most likely source of the infection was contact with harp seals (*Phoca groenlandica*) inhabiting the Arctic sea. Altered migration patterns of Arctic harp seals were noted in the year preceding the PDV epizootic [37,38]. PDV-specific antibodies were detected in 30% (12/40) of archival sera from harp seals collected in Greenland before 1988, but could not be detected in the sera of seals inhabiting the European sea prior to 1988 [39,40]. Contemporaneously with the epizootic in northern Europe, the deaths of approximately 18,000 Baikal seals (*Phoca sibirica*) occurred in Lake Baikal with clinical signs very similar to those of the European seals [41,42]. Subsequent molecular analysis revealed that the cause of death was CDV [43,44,45]. CDV-induced mass mortalities also occurred in Caspian seals (*Phoca caspica*) [46,47].

Cetacean morbilliviruses were first described in harbor porpoises (*Phocoena phocoena*) stranded along the coast of Ireland in 1988. The virus was initially named porpoise morbillivirus (PMV) [48,49]. The same virus was isolated in the Netherlands in 1990 [50]. In the Mediterranean Sea, mass-die-offs of striped dolphins occurred (*Stenella coeruleoalba*) between 1990 and 1992 [51]. This unknown virus spread rapidly from the coast of Spain to the coasts of France, Italy, Greece, and Turkey [4]. Infection was caused by a new virus, which was initially termed the dolphin morbillivirus (DMV) [51,52]. DMV was retrospectively identified as the causative agent of an epizootic in bottlenose dolphins (*Tursiops truncatus*) in the western North Atlantic Ocean in the 1980s [53,54]. PMV and DMV have since been unified as CeMV based on sequence similarity. The origin of CeMV strains remains to be elucidated. Molecular phylogenetic studies have demonstrated that PDV is evolutionarily close to CDV, whereas CeMV is closely related to ruminant viruses, including RPV and PPRV [55,56].

## 3. Epidemiological Studies Based on Molecular Biology and Serology

Since the discovery of these novel marine morbilliviruses, mass-mortality events induced by these viral agents have repeatedly occurred in the same regions or other oceans. In addition to these epizootic studies, stranded whales and pinnipeds have been extensively investigated by serological, pathological, and molecular biology techniques. Recent infections of pinnipeds with PDV and of cetaceans with CeMV are summarized in Table 1 and Table 2, respectively. The sites of reported infections are mapped in Figure 1.

In 2002, a PDV epidemic occurred again along the Northern European coastline killing more than 30,000 seals [60,61]. The harbor seal was the primary species affected by PDV, with grey seals infected to a lesser extent. Both species belong to the family Phocidae within the suborder Pinnipedia. In other locations, infection of pinnipeds with PDV has been reported, including the Phocidae and other families such as the Otariidae and the Odobenidae (Table 1, Figure 1).

After the discovery of CeMV in the Mediterranean Sea in 1990, the virus repeatedly caused outbreaks in 2006–2008, 2011, and 2013 in the same region [96,108]. The main target species were the striped dolphin, the bottlenose dolphin, and the short-finned pilot whale, all of which belong to the family Delphinidae in the suborder Odontoceti. However, in the 2011 and 2013 epizootics infection also spread to the Cuvier’s beaked whale [108], the sperm whale [112], and the fin whale [114,115] (Table 2).

CeMV has also been detected in various cetacean species in oceans across the world (Table 2, Figure 1). Because significant genetic variation has been observed among viral strains, they were initially named based on the cetacean species in which they were first discovered: DMV, PMV, pilot whale morbillivirus (PWMV) [88], beaked whale morbillivirus (BWMV) [94], and Guiana dolphin morbillivirus (GDMV) [106]. Among these viruses, DMV appears to be the most prevalent. Currently all of these strains are thought to represent CeMVs and changing their names to CeMV-1, -2, -3, -4 and -5, respectively, has been proposed [119].

Sirenians, including manatees (*Trichechus manatus*) and dugongs (*Dugong dugon*), are classified within the order Sirenia and the magnorder Afrotheria. They are grass-eating animals that have adapted to an aquatic lifestyle. For the manatee, serological evidence of exposure to morbilliviruses has been reported, although no virus has yet been isolated from these animals [98].

## 4. Pathology and Pathogenesis

Experimental infection of harbor seals with PDV confirmed that its pathological features are similar to those of CDV [120]. Although there are no reports of experimental infection of cetaceans with CeMV, the pathological features of dead cetaceans during mass mortality events resemble those seen for other morbillivirus infections of land animals. Severe bilateral pneumonia is the most common symptom observed, and prominent lymphoid depletion, interstitial pneumonia and meningoencephalitis are also commonly observed [3,4]. Severe immunosuppression promotes opportunistic infections by viruses, bacteria, fungi, and/or protozoa. In histopathology analysis, virus-associated lesions were commonly found in the lung, brain, and lymphoid tissues, and, to a lesser degree, in epithelial and glandular tissues. Cytoplasmic and/or nuclear inclusion bodies and multinucleated giant cells (syncytia) are often observed in infected tissues [3,4].

A crucial contribution by SLAM to the pathogenesis by these viruses has been demonstrated in a model mouse system [121]. Recombinant MVs or CDVs lacking the ability to bind SLAM (SLAM-blind) were not infectious and did not induce clinical symptoms in monkeys or ferrets [22,122]. These observations indicated that SLAM plays an essential role in the pathogenicity of morbilliviruses. The fact that the pathology of morbillivirus-infected marine mammals is similar to that of terrestrial animals indicates that PDV and CeMV also spread in their hosts via their SLAMs. The ability of PDV and CeMV to interact with host SLAMs has been demonstrated recently by in vitro experiments [119,123].

A “brain-only” form of CeMV that causes neuropathogenesis has been described [124]. CeMV antigens were exclusively detected in the brains of a substantial number of CeMV-infected striped dolphins in the Mediterranean Sea [125,126,127]. This suggests that dolphins develop chronic encephalitis after recovery from an acute viral systemic infection. The lesions observed in these animals were highly similar to subacute sclerosing panencephalitis (SSPE) observed in MV patients [128,129]. Histochemical studies have suggested that spread of CeMV occurs via cell-to-cell transmission in dolphin brains and is similar to that found in SSPE in humans [125,126,127]. SSPE is rare and usually occurs in MV-infected children at an incidence of eight to 20 cases per million several years after recovery from acute measles [128,129]. However, a recent study showed a high incidence of SSPE (one SSPE case per 609 acute measles cases) in children <12 months of age [130]. The “brain-only” form of CeMV was observed in eight of 44 CeMV-positive striped dolphins found stranded in the Mediterranean between 1990 and 1995. Although the incidence of SSPE and CeMV-induced neuropathogenesis cannot be directly compared, “brain-only” CeMV still warrants careful attention [131]. MV strains responsible for SSPE have been reported to possess a number of mutations, mainly in the M gene and, less frequently, in the F and H genes [132,133]. Molecular analyses of “brain- only” CeMV should contribute to an understanding of its pathology.

There have been some reports suggesting that subclinical CeMV infection can occur in cetaceans, with viral RNA or specific antibodies detectable in the absence of histological lesions [79,89]. The absence of clinical signs may mask the infection, leading to an underestimation of incidence or prevalence. Healthy-seeming infected animals may still be at increased risk of opportunistic or secondary infections due to immunosuppression, but may also be protected against more virulent strains. Accumulation of more detailed information on subclinical infections will be important for understanding the epidemiology of CeMV and for protecting marine mammals.

## 5. Transmission and Maintenance of PDV and CeMV

Morbilliviruses never survive long in the environment outside their hosts. They are transmitted from acutely infected animals to susceptible ones by close contact or via airborne droplets. Seropositivity of CeMV-specific antibodies was reported to decrease from 100% to 50% from 1990–1992 to 1997–1999 in mature striped dolphins located in the Mediterranean Sea. This result suggests that CeMV does not persist in the dolphin population [80]. In general, morbillivirus transmission occurs mainly by inhalation of aerosolized virus particles released from infected animals. Inoculation experiments of monkeys and ferrets by the intranasal route using green fluorescent protein-expressing MV and CDV, respectively, have shown that the viruses first appear in macrophages or dendritic cells at the alveolar level, replicate in lymphocytes, and then spread to lymphoid tissues by cell-associated viremia [24,134,135]. The viruses finally reach the airway epithelial tissue, replicate in high numbers, and are released into the respiratory tract and air [20,23,26,136,137,138]. Contact transmission is also thought to occur. Recently, biting by ferrets has been shown experimentally to be a route of CDV infection via nectin-4 by use of SLAM-blind CDV [28].

Pinnipeds often form densely populated colonies during their land life and may come in close contact with infected animals. Thus, PDV may be transmitted in a similar manner to that observed for terrestrial animals. Although life history and behavior are not fully understood in cetaceans, breathing in a synchronized manner may facilitate aerosol transmission. Maternal transmission has also been postulated for CeMV transmission [99,124]. Viral RNA was detected in various tissues of a seven-month fetus of a DMV-infected long-finned pilot whale, reflecting vertical transmission [99]. Collection of related information and new research is necessary for understanding vertical transmission of these viruses.

Because infection by morbilliviruses induces lifelong immunity, new susceptible animals must constantly be supplied for maintenance of the virus in the population. It has been calculated that the minimal population size for MV maintenance is approximately 300,000 individuals [139]. If a certain morbillivirus possesses an H protein that binds efficiently to a broad range of SLAMs, the virus can possibly be shared among different animal species, which would ensure that the morbillivirus is maintained in an animal community with an expanded population size.

## 6. Interaction between SLAM and the Viral H Protein

### 6.1. Phylogenetic Relationship between SLAMs and Viral H Proteins

In contrast to nectin-4, whose sequence is highly conserved across species [31,32], SLAM has significant sequence diversity among host animals and is thought to contribute to the tropism of morbilliviruses [29,30]. We present phylogenetic trees of the deduced amino acid sequences of SLAMs and morbillivirus H proteins (Figure 2). The topology of the phylogenetic tree of SLAMs is consistent with that of the generally accepted mammalian phylogenetic tree (Figure 2a) [140,141]. Ruminants (A and B in Figure 2a) and cetaceans (C) form one group, corresponding to the order Cetartiodactyla. Pinnipeds (F) form an additional group along with dogs (E) and cats (D), corresponding to the order Carnivora. These two groups constitute a larger clade, corresponding to the upper-order Laurasiatheria. Primates (G) and manatees and elephants (H) form independent clades, which belong to the upper-order Euarchontoglires and the magnorder Afrotheria, respectively. The latter was used as an outgroup for the tree (Figure 2a). The morbillivirus phylogeny based on full-length MV H protein sequences reflects the host taxonomic grouping, except for MV (Figure 2b). MV (G) forms a robust clade with two ruminant viruses (A and B). These phylogenetic analyses strongly suggest that SLAMs and viral H proteins have co-evolved. However, host switching seems to have occurred in the evolution of MV, which might have originated from an ancestral ruminant virus by acquiring binding affinity for human SLAM [30].

### 6.2. The SLAM–Viral H Protein Interface

SLAM is a type I transmembrane protein belonging to the SLAM family of the immunoglobulin superfamily. The extracellular region of this protein is composed of a membrane–distal immunoglobulin variable (V) domain and a membrane–proximal immunoglobulin constant (C2) domain [144]. SLAM is a self-ligand and binds to the V domain of another SLAM on a different cell with weak affinity *in vivo* [145]. The V domain provides an interface for the morbillivirus H protein [146]. Because morbillivirus H protein binds to a similar region of the V domain with 400-fold higher affinity than the SLAM–SLAM interaction, replacement of a bound SLAM molecule with the H-protein is thought to occur readily [147]. Crystal structures of the complex between MV-H and cotton-top tamarin (*Saguinus oedipus*) SLAM-V demonstrated that the viral H protein contains a six-bladed β-propeller fold in the head domain and interacts with the SLAM-V domain at the β 4–6 region [148]. The cotton-top tamarin SLAM was used in this study because B95a cells, which are highly susceptible to wild-type MV strains, are derived from cotton-top tamarin B lymphocytes [149]. Four main binding sites contribute to the interaction between these two proteins [148]. To identify the interfaces of marine mammal SLAMs, three-dimensional homology models were constructed based on the crystal structure of the MV H protein–SLAM complex [150,151]. The SLAM interfaces of killer whales (*Orcinus orca*) and cotton-top tamarins are shown in Figure 3. The front β-sheet structure of the SLAM-V domain contains four anti-parallel β-strands and provides an interface for the morbillivirus H protein. The residues at this interface are involved in H protein binding (Figure 3b,c).

### 6.3. SLAM Residues Important for Binding to the Morbillivirus H Protein

The amino acid side chains that protrude from the SLAM interface are likely to be involved in binding to the viral H protein (Figure 3b,c). In addition, specific residues of SLAM involved in binding to the viral H protein have been identified through structural and virological studies [148,152,153]. In total, 35 residues are important for this interaction (Table 3). Approximately half of these 35 residues are conserved among animal species that are known hosts of morbilliviruses, thus appear to be crucial for binding to the viral H protein as well as for SLAM function as an immune molecule. The remaining half, which are heterogeneous among host animals, may also be important for binding the H protein and contribute to host–virus specificity. In particular, positions occupied by residues that have different charges (residues 68, 71, 72, 75, 76, 84, 85, 87, 90, 124, 129, and 130) may be important for modulating the affinity of the SLAM–H protein interaction and also for determining the host specificity of the virus (Table 3).

### 6.4. Sequence Variations in the 35 SLAM Residues Potentially Involved in H Protein Binding among Pinnipeds and Carnivores

Alignment of the SLAM V domain amino acid sequences of pinnipeds and other carnivores is shown in Appendix A. The 35-residue SLAM subsequences from representative animals were also used to construct a mammalian phylogenetic tree (Figure 4). Order Carnivora is largely classified into two suborders: the Caniformia including pinnipeds and dogs, and the Feliformia, which includes lions and cats. Pinnipeds are composed of three families: the Phocidae, Odobenidae, and Otariidae. The 35-residue SLAM subsequences were identical in all three families of pinnipeds except for a single residue substitution in the northern fur seal (*Callorhinus ursinus*; Appendix A). The 35-residue SLAM subsequences of pinnipeds showed only one to three substitutions when compared with other members of the Caniformia (Figure 4, Appendix A). This is consistent with previous reports of PDV infections in otters and minks [154,155], and with CDV-induced mass mortality events in Baikal and Caspian seals [156]. Various wild Caniform animals such as foxes, raccoons, and otters are also infected with CDV [157,158]. PDV and CDV are closely related viruses and the SLAM sequences of Caniform animals are similar (Figure 2, Appendix A). If these animals come into contact in nature, morbilliviruses may be transmitted between them, potentially representing a single multispecies target community.

Outbreaks of CDV infection have occurred in large cats such as lions and tigers, reflecting host expansion of CDV [163,164]. Nine of the 35 SLAM subsequence residues differ between dog and lion SLAMs, and charge differences occur at three positions: 72, 82, and 129 (Figure 4, Appendix A) [165]. In particular, K72 and S82 are unique to large and small cats among host mammals. This difference may reduce or eliminate the ability of CDV to naturally infect the Feliformia, although it is not critical for interaction of CDV with cat SLAM *in vitro* [166]. The H protein of a lion CDV strain was shown to potentially infect a broad range of carnivores with moderate affinity, and unique residues of the lion CDV-H protein play a crucial role in infection [167,168]. Between large and domestic cats, only one substitution involving charge alteration is present at residue 76 (Figure 4, Appendix A) [165]. It remains unclear whether this difference produces a major barrier for dog and/or lion CDVs to infect domestic cats. Nikolin et al. [167] proposed a hypothesis of “specialist” and “generalist” CDVs. The “specialist” CDV has strong affinity toward a narrow range of hosts, whereas the “generalist” CDV possesses moderate affinity toward a broader range of hosts. Which strategy type is more advantageous for survival seems to largely depend on ecological and environmental situations. In any case, it is probably advantageous for most viruses to fall between these two extremes. Morbilliviruses that infect wild animals may have evolved by choosing one of these two contradictory strategies (i.e., expansion and specialization).

### 6.5. Sequence Variations in the 35 SLAM Residues Potentially Involved in H Protein Binding Among Cetaceans

An alignment of SLAM V domain sequences from various cetaceans is shown in Appendix A and Figure 4. The 35-residue SLAM subsequence involved in binding the H protein is more variable among cetacean species than pinnipeds [150]. For example, between striped dolphin and fin whale, six of the 35 residues differ, including three substitutions with charge differences at positions 68, 90, and 130 (Figure 4, Appendix A). During the mass mortality events in the Mediterranean Sea, the prevalent DMV strain (a major CeMV strain) infected a broad range of cetacean species, including the striped dolphin, the Cuvier’s beaked whale (*Ziphius cavirostris*), the sperm whale, and the fin whale [51,108,112,114,115]. This indicates that the CeMV H protein likely has weaker binding specificity, enabling this protein to interact with a range of cetacean SLAMs. This somewhat promiscuous binding activity would enable the virus to infect multiple cetacean species, which would dramatically increase the population of susceptible animals, even though the population of each individual species may be quite small. Interestingly, the SLAM 35-residue subsequence for fin whale has a higher sequence homology to that of cows than striped dolphins; polymorphisms occur at three residues with two changes of charged amino acids (Figure 4, Appendix A).

### 6.6. Host Switching and Expansion of CeMV Beyond the Order

Each morbillivirus usually infects animals within a single order. However, cases of host switching or expansion beyond an order have been reported. Phylogenetic analysis of morbilliviruses strongly suggested that host switching occurred from ruminants to humans during the evolution of MV from an ancestral ruminant morbillivirus (Figure 2) [30,169]. Figure 5 summarizes the relationships between viral species and the order-level host taxonomy along with selected residues from the 35-residue SLAM subsequence critical for H protein binding. To highlight host specificity, only residues not conserved among host animals are shown.

CeMV was transmitted to a captive harbor seal from rescued bottlenose dolphins during an epidemic that occurred in the Mediterranean Sea, and the harbor seal died from the infection [170]. CeMV infection has also been reported previously in Mediterranean monk seals (*Monachus monachus*) [171,172]. In *in vitro* experiments, CeMV has been shown to interact efficiently with seal and dog SLAMs [119,173,174]. Therefore, for CeMV transmission to seals, the barrier effect of SLAM appears to be low even though the SLAM interfaces with viral H protein are dissimilar: Eleven amino acid differences are present in the 35-residue SLAM subsequence critical for H protein binding between bottlenose dolphins and harbor seals, seven of which involve charge differences (Table 3, Figure 4, Appendix A). Natural infection of seals with CeMV; however, has rarely been documented. This suggests that host factors other than SLAMs are involved in the host–virus specificity of morbilliviruses [175,176]. In addition, ecological factors that facilitate contact of host animals may be a key factor in determining morbillivirus host specificity.

Recently, CDV has been reported to cause fatal infections in monkeys of the *Macaca* genus (Figure 5) [177,178,179]. Interestingly, CDV can use the *Macaca* but not the human SLAM even though the amino acid sequences of their SLAM interfaces are identical. After acquiring any of several single residue substitution (R519S, D540G, and P541S) on the CDV H protein, the CDV readily adapted to recognition of human SLAM [180]. Single residue substitutions in the PPRV H protein conferred the ability to recognize human SLAM in *in vitro* experiments [181]. These results indicate that a single residue substitution of the H protein can alter the interaction with human SLAM and that even an incongruent complex (e.g., CDV and human SLAM) can induce a subtle conformational shift that can trigger fusion. Furthermore, these results indicate that residues outside the SLAM–H protein interface are also involved in this process [180].

Although the precise mechanisms involved in barriers to host specificity of morbilliviruses remain to be elucidated, several factors are thought to be involved. Ecological and host factors such as innate immune factors may play important roles. However, SLAM still seems to function as a host barrier. The H proteins of MV vaccine strains are able to bind three different receptors (SLAM, nectin-4, and CD46) via the same region of the molecule, a β4-β5 groove using different residues [148,182,183]. This demonstrates that the binding region of the H protein contains at least three closely spaced but specific interfaces that trigger fusion. To further understand host–virus specificity, the finely-tuned interaction between the viral H protein and its receptors, as well as any resulting conformational changes, must be studied by structural analyses.

For host range expansion beyond hosts within a single taxonomic order, the viral H protein most likely adopts a “generalist” approach to increase the number of binding partners. Morbilliviruses in humans and domesticated animals may have evolved to become “specialists” as targets for infection are amply available. In contrast, the “generalist” strategy among virus populations may play an active role in host switching or expansion. Progress in recent veterinary research on wild animals is indicating a role for a “generalist” strategy with more divergent interactions between host animals and viruses [167].

## 7. In silico Analysis

Crystal structural analyses have shown that the wild-type MV-H protein binds to SLAM and nectin-4 via the same region, the β4-β5 groove, although different amino acids are involved in each interaction [148,183]. The importance of some amino acids has been shown by virological *in vitro* experiments. However, detailed interactions between residues of MV-H and host receptors remain largely unexplored. This problem has recently been addressed by calculating inter-fragment interaction energies in complexes of MV-H and SLAM-V or nectin-4 using a fragment molecular orbital (FMO) method [184].

This analysis revealed that ion pair contributions were more frequent in the MV-H–SLAM complex than in the MV-H–nectin-4 complex, in which the contribution of hydrophobic interactions was more prominent [184]. In the MV-H–SLAM complex, seven key residues of MV-H with the highest binding energies were charged residues interacting with oppositely-charged SLAM residues. Two charged SLAM residues (K77 and E123), which are conserved in all reported SLAMs (Figure 4), play a key role in binding [184]. These calculations are consistent with previous virological and structural studies. Furthermore, in silico analyses have quantitatively examined the complex nature of the interactions between the H protein and its receptors. A single residue on the H protein interacts with multiple residues of the corresponding receptor and vice versa [184]. This also indicates that residues outside the interface may affect complex formation and stability. Systematic computational analyses will provide a useful tool for understanding the interactions between morbilliviruses and host cell receptors, as well as viral host expansion or alteration over evolutionary time.

## 8. Conclusions

Reports of infection of pinnipeds and cetaceans in various oceans with PDV and CeMV have continued to emerge. Infection induces an acute lymphotropic disease in marine mammals. SLAM is a receptor responsible for viral entry and pathogenesis and plays an important role in determining host specificity [16,22,121,122]. Based on the crystal structures and homology models of SLAMs, 35 SLAM residues are seemingly important for interaction with the morbillivirus H protein and form the binding interface [30,151]. These 35 residues are essentially conserved among the pinnipeds and share high similarity among members of the suborder *Caniformia*, including dogs, otters, and mink. Consequently, PDV and CDV are postulated to infect these animals [165], forming a single multispecies mammalian community of hosts for the two viruses. This set of 35 critical SLAM residues is more divergent among cetaceans [150]. Because CeMV infects a broad range of cetacean species, its H protein may have broader binding specificity to facilitate interaction with more diverse cetacean SLAM sequences. This indicates that CeMV may target multiple cetacean species as hosts. Recently, it has been reported that seals were infected with CeMV, and that CeMV interacted with seal SLAM in *in vitro* experiments [119,170]. These observations indicate that CeMV can infect seals, although SLAM still seems to contribute to host–virus specificity in nature. Recently, host expansion of CDV to monkeys of the *Macaca* genus but not to humans has been reported, even though both SLAM interfaces are identical [179]. A single residue substitution of the CDV H protein sequence appears to be responsible for the selective interaction with human SLAM [180]. To understand host switching or expansion beyond members of a single order, additional structural studies of complexes between various host and viral molecules, including apparently incongruent combinations of SLAMs and H proteins in various conformations, will be required. To understand the detailed protein–protein interactions between the interfaces of the ligand–receptor complex, computational analysis using a FMO method has been applied [184]. This study demonstrated that a quantitative evaluation of complex residue–residue interactions is possible. Host switching or expansion beyond a single order also suggests the existence of other important factors that modulate the host–virus specificity of morbilliviruses. It is important to identify novel host factors that interact with viral components. Comprehensive studies involving epidemiological, virological, structural, and systematic computational analyses should provide a deeper understanding of host–virus specificity in the near future.

## Figures and Tables

**Figure 1 viruses-11-00606-f001:**
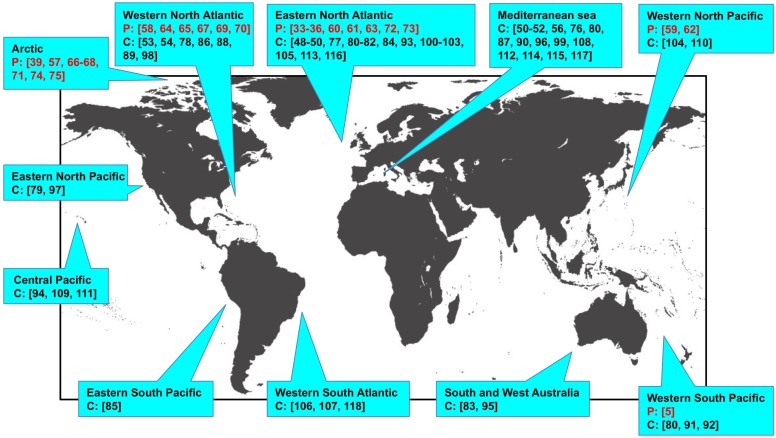
Locations of reported phocine distemper virus infections of pinnipeds and cetacean morbillivirus infections of cetaceans. Numbers indicate references. P (red), pinnipeds; C (black), cetaceans.

**Figure 2 viruses-11-00606-f002:**
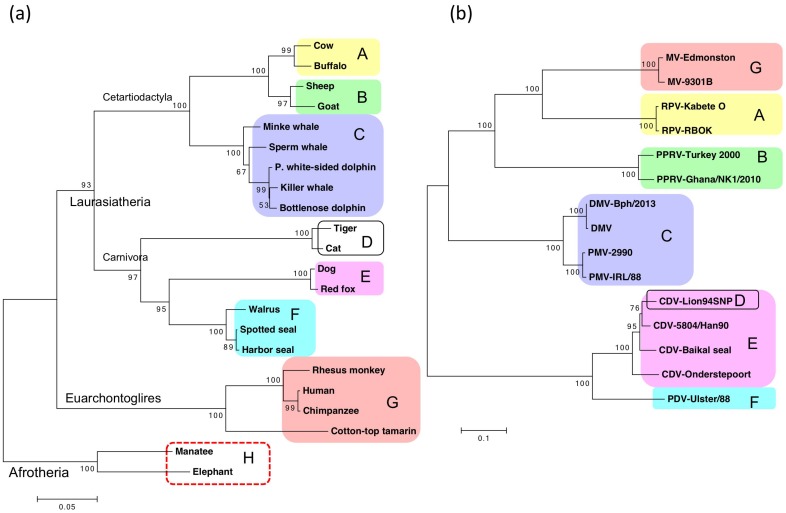
Phylogenetic trees of full-length signaling lymphocyte activation molecules (SLAMs) (**a**) and full-length morbillivirus hemagglutinin (H) proteins (**b**). Neighbor-joining methods were implemented in MEGA7 [142,143]. The bootstrap probabilities with 500 replications are shown at the nodes. Respective pairs of host animals and corresponding morbilliviruses are shown as squares with the same color and same alphabetic notations: A, large ruminants; B, small ruminants; C, cetaceans; D, felids; E, canids; F, pinnipeds; G, primates. Morbilliviruses have not been identified in manatees and elephants (H, red dashed box). The amino acid sequences of full-length SLAMs were obtained from the Data Bank of Japan: cow (*Bos taurus*, AAK61860.1); buffalo (*Bubalus bubalis*, ABB58751.1); sheep (*Ovis aries*, NP_001035378.1); goat (*Capra hircus*, ABB58752.1); minke whale (*Balaenoptera acutorostrata*, XP_007171815.1); sperm whale (*Physeter catodon*, XP_007124119.2); Pacific white-sided dolphin (*Lagenorhynchus obliquidens*, BAH10670.1); killer whale (*Orcinus orca*, BAH1067.1); bottlenose dolphin (*Tursiops truncatus*, XP_004327894.1), tiger (*Panthera tigris altaica*, XP_007092436.1); cat (*Felis catus*, BAN42597.1); dog (*Canis lupus familiaris*, AAK61857.1); red fox (*Vulpes vulpes*, ACD47119.1); walrus (*Odobenus rosmarus*, BAH10673.1); spotted seal (*Phoca largha*, BAH10672.1); harbor seal (*Phoca vitulina*, AYR16904.1); rhesus monkey (*Macaca mulatta*, XP_001117605.1); human beings (*Homo sapiens,* NP_003028.1); chimpanzee (*Pan troglodytes*, XP_513924.2); cotton-top tamarin (*Saguinus oedipus*, AAG02017.1); manatee (*Trichechus manatus*, BAH10674.1); and elephant (*Loxodonta africana*, XP_003415237.1). The amino acid sequences of morbillivirus full-length H proteins were obtained as follows: MV Edmonston AIK-C strain (BAB60866.1); MV 9301B strain (BAA33872.1); RPV Kabete O strain (AAA47401.1); RPV RBOK strain (CAA83182.1); PPRV Turkey 2000 strain (YP_133827.2); PPRV Ghana/NK1/2010 strain (AID07002.1); DMV Bph/2013 strain (AOZ56997.1); DMV (NP_945029.1); PMV 2990 strain (AAT84062.1); PMV IRL/88 strain (ACV50419.1); CDV Lion94SNP strain (AFQ38775.1); CDV 5804/Han90 strain (CAA59359.1); CDV Baikal seal strain (CAA59357.1); CDV Onderstepoort vaccine strain (AAG30920.1); and PDV Ulster/88 strain (BAA01207.1). Each amino acid sequence alignment was generated using ClustalW version 1.8.

**Figure 3 viruses-11-00606-f003:**
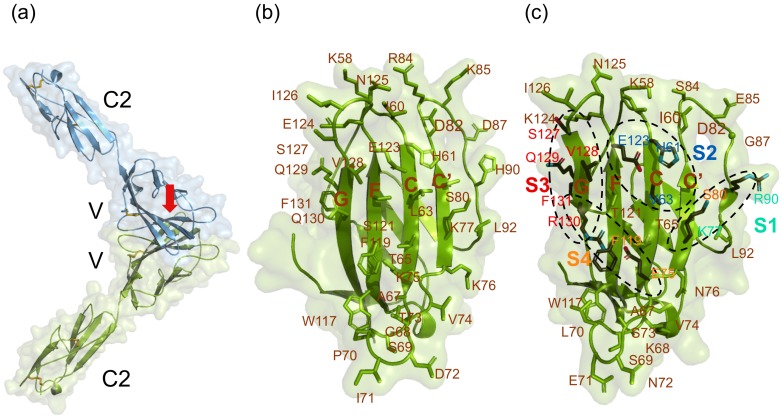
Ribbon representation of the three-dimensional (3D) structures of signaling lymphocyte activation molecules (SLAMs) from killer whales (*Orcinus orca*) and cotton-top tamarins (*Saguinus oedipus*). The 3D models were constructed by homology modeling based on the crystal structure of the complex between MV-H and the cotton-top tamarin SLAM-V [148,150]. (**a**) Blue and green models show the two SLAM extracellular domains of killer whales forming a homophilic dimer. β-strands are indicated by blue and green arrows and disulfide bonds are shown as yellow lines. The red arrow indicates the direction of view of the interface for morbillivirus binding as shown in (**b**) and (**c**). (**b**) Interface of the killer whale SLAM from the view of the red arrow in (**a**). Four β-strands are indicated as C, C’, F, and G. The amino acids that potentially interact with the viral H protein are shown along with their position numbers. The amino acid positions correspond to those of the MV-H–SLAM-V crystal structure [148]. (**c**) As a reference, the cotton-top tamarin SLAM interface is shown. The four binding sites (S1–S4) present in the crystal structure [148] are marked with dotted circles. The residues involved in these four binding sites are colored differently (site 1—mint green; site 2—cyan; site 3—red; site 4—orange) with colored atoms (black for carbon, blue for nitrogen, and red for oxygen). This figure was modified from Shimizu et al. [150].

**Figure 4 viruses-11-00606-f004:**
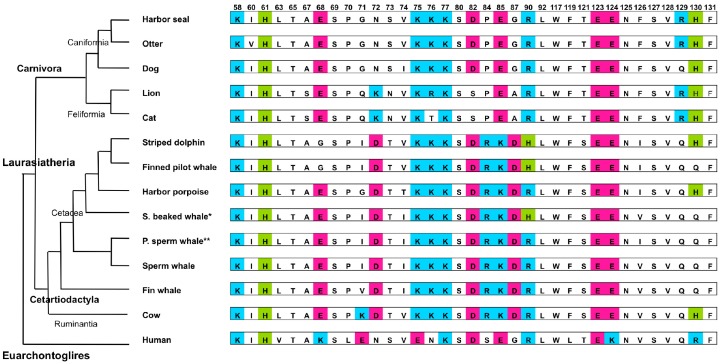
Alignment of the 35-residue signaling lymphocyte activation molecule (SLAM) subsequence important for interaction with the hemagglutinin (H) protein along with the corresponding phylogenetic tree. The mammalian phylogenetic tree was created by merging a conventional mammalian tree [140,141], the detailed cetacean tree generated using the short interspersed elements method [159,160,161], and the detailed tree of Caniformia generated using maximum likelihoods method [162]. Cyan, green, and pink boxes, respectively, indicate positively, weak-positively, and negatively charged amino acids. * Stejneger’s beaked whale, ** Pygmy sperm whale.

**Figure 5 viruses-11-00606-f005:**
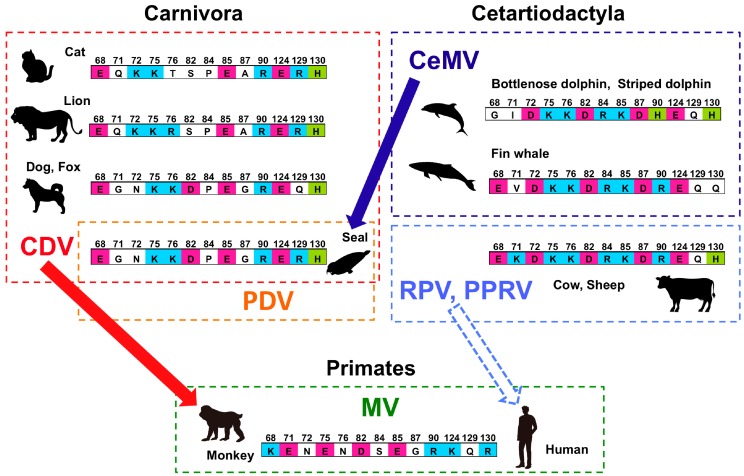
Schematic diagram of the relationships between morbillivirus species and their host animals. The hosts of each morbillivirus species are surrounded by dotted squares. The names of the various morbilliviruses are given in different colors. Residues of the 35-residue signaling lymphocyte activation molecule (SLAM) subsequence involved in hemagglutinin (H) protein binding that are not conserved among the host animal species are shown. Conserved residues or conservative amino acid substitutions are not shown. Thick colored arrows indicate recent host expansion beyond the order, and the dotted arrow indicates that the inferred transmission event during the evolution of MV.

**Table 1 viruses-11-00606-t001:** Pinniped species infected with phocine distemper virus.

Family	Species	References
Phocidae	Harbor seal (*Phoca vitulina*)	[33] *, [34,35,36,57,58,59], [60,61] *, [62], [63,64,65] *
	Ringed seal (*Phoca hispida*)	[39,66,67]
	Spotted seal (*Phoca larga*)	[59]
	Harp seal (*Phoca groenlandicus*)	[39], [65] *, [66,67,68,69,70]
	Grey seal (*Halichoerus grypus*)	[57,58,65,71,72,73]
	Hooded seal (*Cystophora cristata*)	[66,67,70]
Odobenidae	Walrus (*Odobenus rosmarus*)	[74,75]
Otariidae	Steller sea lion (*Eumetopias jubatus*)	[59]
	Hooker’s sea lion (*Phocartos hookeri*)	[5]
	New Zealand fur seal (*Arctocephalus forsteri*)	[5]

* Virus isolation and/or polymerase chain reaction analysis.

**Table 2 viruses-11-00606-t002:** Cetacean species infected with cetacean morbillivirus.

Family	Species	References
Odontoceti		
Delphinidae	Common dolphin (*Delphinus delphis*)	[76,77,78], [79] *, [80], [81,82,83] *, [84]
	Long-beaked common dolphin (*Delphinus capensis*)	[85]
	Bottlenose dolphin (*Tursiops truncatus*)	[53], [54] *, [80], [83] *, [85,86,87], [88] *, [89], [90,91,92,93,94] *
	Indo-Ocean bottlenose dolphin (*Tursiops aduncus*)	[83,92,95] *
	Striped dolphin (*Stenella coeruleoalba*)	[51,52] *, [76,78,80], [81,82,90,94,96,97] *
	Atlantic spotted dolphin (*Stenella frontalis*)	[78]
	Long-finned pilot whale (*Globicephala melas*)	[80], [88,90] *, [98], [99,100] *
	Short-finned pilot whale (*Globicephala macrorhynchus*)	[81,98], [100,101] *
	White-beaked dolphin (*Lagenorhynchus albirostris*	[50,77], [102,103] *
	Atlantic white-sided dolphin (*Lagenorhynchus acutus*)	[78]
	Pacific white-sided dolphin (*Lagenorhynchus obliquidens*)	[104]
	Dusky dolphin (*Lagenorhynchus obscurus*)	[85]
	Rough-toothed dolphin (*Steno bredanensis*)	[94] *
	Spotted dolphin (*Stenella attenuata*)	[94] *
	Spinner dolphin (*Stenella longirostris*)	[94] *
	Fraser’s dolphin (*Lagenodelphis hosei*)	[78,80,92]
	Risso’s dolphin (*Grampus griseus*)	[80], [94,105] *
	False killer whale (*Pseudorca crassidens*)	[78]
	Melon-headed whale (*Peponocephala electra*)	[92]
	Pygmy killer whale (*Feresa attenuata*)	[78]
	Guiana dolphin (*Sotalia guianensis*)	[106,107] *
Phocoenidae	Harbor porpoise (*Phocoena phocoena*)	[48,49,50] *, [77,78,80]
Ziphiidae	Cuvier’s beaked whale (*Ziphius cavirostris*)	[94,108] *
	Longman’s beaked whale (*Indopacetus pacificus*)	[94,109] *
	Blainville’s beaked whale (*Mesoplodon densirostris*)	[94] *
Kogiidae	Pygmy sperm whale (*Kogia breviceps*)	[78], [94,110] *
Physeteridae	Sperm whale (*Physeter macrocephalus*)	[94,111,112] *
Mysticeti		
Balaenopteridae	Fin whale (*Balaenoptera physalus*)	[56] *, [113], [114,115,116] *
	Common minke whale (*Balaenoptera acutorostrata*)	[117]
	Bryde’s whale (*Balaenoptera edeni*)	[92]
	Humpback whale (*Megaptera novaeangliae*)	[94] *
Balaenidae	Southern right whale (*Eubalaena australis*)	[118] *

* Virus isolation and/or polymerase chain reaction analysis.

**Table 3 viruses-11-00606-t003:** Comparison of 35 residues at the signaling lymphocyte activation molecule (SLAM) interface potentially involved in binding to the hemagglutinin (H) proteins of the six main morbillivirus host animals.

**(a)**																				
**Residue ^1^**	**58**	**60**	**61**	**63**	**65**	**67**	**68**	**69**	**70**	**71**	**72**	**73**	**74**	**75**	**76**	**77**	**80**	**82**	**84**	**Virus**
**β-strand ^2^**			**C**	**C**	**C**	**C**									**C’**	**C’**	**C’**			
**Site ^3^**			**2**	**2**										**4**		**1**				
Seal	K	I	H	L	T	A	E	S	P	G	N	S	V	K	K	K	S	D	P	PDV
Dog	K	I	H	L	T	A	E	S	P	G	N	S	I	K	K	K	S	D	P	CDV
Dolphin	K	I	H	L	T	A	G	S	P	I	D	T	V	K	K	K	S	D	R	CeMV
Cow	K	I	H	L	T	A	E	S	P	K	D	T	V	K	K	K	S	D	R	RPV
Sheep	K	I	H	L	T	A	E	S	P	R	D	T	V	K	K	K	S	D	R	PPRV
Humans	K	I	H	V	T	A	K	S	L	E	N	S	V	E	N	K	S	D	S	MV
**(b)**																				
**Residue ^1^**	**85**	**87**	**90**	**92**	**117**	**119**	**121**	**123**	**124**	**125**	**126**	**127**	**128**	**129**	**130**	**131**	**Virus**			
**β****-strand ^2^**					**F**	**F**	**F**	**F**				**G**	**G**	**G**	**G**	**G**				
**Site ^3^**			**1**			**4**		**2**				**3**	**3**	**3**	**3&4**	**3**				
Seal	E	G	R	L	W	F	T	E	E	N	F	S	V	R	H	F	PDV			
Dog	E	G	R	L	W	F	T	E	E	N	F	S	V	Q	H	F	CDV			
Dolphin	K	D	H	L	W	F	S	E	E	N	I	S	V	Q	H	F	CeMV			
Cow	K	D	R	L	W	F	S	E	E	N	V	S	V	Q	H	F	RPV			
Sheep	K	G	H	L	W	F	S	E	E	N	V	S	V	Q	H	F	PPRV			
Humans	E	G	R	L	W	L	T	E	K	N	V	S	V	Q	R	F	MV			

The 35 residues were extracted from the following sequences of full-length SLAMs: harbor seal (*Phoca vitulina*, AYR16904.1); dog (*Canis lupus familiaris*, AAK61857.1); striped dolphin (*Stenella coeruleoalba*, BAN84418.1); cow (*Bos taurus*, AAK61860.1); sheep (*Ovis aries*, NP_001035378.1); and humans (*Homo sapiens*, NP_003028.1). ^1^ The residue number, ^2^ β-strand symbols, and ^3^ the site name of the four binding sites are based on those shown in the crystal structure (PDB ID: 3ALZ) of the complex between MV-H and cotton-top tamarin SLAM-V [148]. Blue, green, and pink boxes indicate residues with positive charge, weak positive charge, and negative charge, respectively.

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
