# Peer review of "Marine Morbilliviruses: Diversity and Interaction with Signaling Lymphocyte Activation Molecules"

_viruses, 2019, doi:10.3390/v11070606_

Reviewer 1 Report

In general, the manuscript is well structured and presented especially with respect to the figures but English usage needs to be improved. Sections 1 – 5 recapitulate subjects that have already been extensively reviewed elsewhere but section 6 provides new and very useful analysis of CD150 sequences. Whereas extensive clinical, epidemiological and more recently sequencing data has been previously published for both PDV and CeMV, deeper analyses based on real laboratory experiments examining receptor usage rather than in silico predictions based on homology models are lacking in this field. However, the comprehensive and detailed analysis of the available sequence data will be very useful for researchers in the field and provides a roadmap towards such in vitro mechanistic experiments.

Major comments

(1)    Tables 1 and 2: In many of the listed studies, evidence for infection is indirect due to positivity of serum samples for antibodies which recognise PDV or CeMV. It would be very useful for the reader if these studies could be differentiated from those studies reporting RT-PCR or virus isolation data by simply putting an asterisk beside those only reporting serology data.

(2)    Please check English usage throughout the manuscript as there are many sentences throughout which could be improved to improve sense and readability (a small number of examples are listed below)

(3)    Lines 255-256: There is no published evidence to suggest that SLAM and morbillivirus H proteins have co-evolved, especially considering that infections in many species are relatively recent

Minor comments

(1)    Abstract, line 15, Should read ‘since their discovery thirty years ago’

(2)    Introduction, lines 33-34: please check English usage

(3)    Line 34: Five morbilliviruses have been described in terrestrial animals – please add feline morbillivirus.

(4)    Lines 35-36: please remove ‘the’ from before each virus name and perhaps also use the new ICTV terminology for these morbilliviruses in brackets after each virus name

(5)    Line 92: Please change this title, as written it makes no sense

(6)    Line 100: ‘were noted in the year’ – which year? 1988 or the year preceding the PDV outbreak?

(7)    Line 194-196: Of course, the listed numbers do not include dolphins which recovered from CeMV infection so the rate of persistent CeMV infection of the CNS is probably a lot lower

(8)    Line 195: Please correct the indicated dates

Author Response

Author’s reply to Reviewer 1

Authors thank the reviewer 1 for many valuable comments. Because  he asked us to improve English usage (major comment (2) and minor comment (2)) of the mansucript, which had been edited by an English editing company, our manuscript was edited again by a professional English editing company. The major revised or modified regions corresponding to the major comment (3) and the minor comments, except for the English usage, are written in red in the revised manuscript. Line numbers expressed in this reply are those in the revised version.

Major comments

(1) According to the comment, we marked the studies reporting RT-PCR or virus isolation with an asterisk in Table 1 and Table 2. The explanation for the asterisk was given in a footnote of each Table. We found some mistakes in the tables in the previous version and corrected them in the revised version. 

(2) We believe that English usage in our manuscript has been improved by the second editing of the professional company. Because the titles of supplemental data were improved, the supplemental data were also uploaded together with the text.

(3) Because the topologies of phylogenetic trees based on the SLAM and H protein were congruent except the positions of the human SLAM and MV, we think the result strongly suggests the co-evolution of the viruses and host animals except human and measles virus. We changed the word, “indicate”, to “strongly suggest” (red in line 262). The coevoluton is the result of the viral specificities against the host animals. The incongruency of the positions of human SLAM and measles virus was explained by a host switching. The unexpected infection events beyond the order that have been reported recently, may be related to host switching. We have discussed this issue in the manuscript (6.6).

Minor comments

(1) We corrected “20” to “30” (red in line 16).

(2)The sentence was improved (red in line 33-34).

(3) Newly discovered feline morbillivirus has a different characteristics from those of other classic morbilliviruses. Its receptor usage and the relation between the receptor and pathogenesis are still not clear. This is the reason why we think that it is appropriate not to describe the feline morbillivirus in the manuscript. For describing this restriction (exclusion of the feline morbillivirus), we used the term, “classic morbilliviruses”, for MV, RPV and PPRV and CDV in the text (red in line 34). This term, does not include the feline morbillivirus. 

(4) According to the comments, we deleted “the” from before each virus name and added the new ICTV terminology in brackets after each virus name (red in line 35-40). Because new term of cetacean morbillivirus is the same as old one, it was not repeatedly described in a bracket.

(5) We changed the title to “Discoveries of marine morbilliviruses” (red in line 93).

(6) We corrected the phrase to “the year preceding the PDV epizootic” (red in line 100-101).

(7) As the reviewer has pointed out, recovered dolphins were not included in the number. We changed the sentence (red in line 199-201). 

(8) We corrected to “1990-1995” (red in line 199).

 Reviewer 2 Report

Dear Editor

I found this review developed in 8 chapters very interesting and well constructed. The figures are clear and well described. In a context of a persistent circulation of human measles virus in both low income and high income countries, it is important to increase knowledge about other morbillivirus and their capacities to switch between species. The authors focused on the interaction of SLAM and a 35 aa fragment of the H protein. Their is no description of other interactions, like fusion viral protein and the cell. That's why I will suggest that the title of this review should be adapted. 

I recommend the publication of this review. Nevertheless, my competences in the domain of marine morbilliviruses are limited. 

Author Response

Author’s reply to Reviewer 2

According to the comment, we changed the title of our manuscript to “Marine Morbilliviruses: Their Diversity and Interaction with Signaling Lymphocyte Activation Molecules” (red in line 2-3).